# ATRD: A Comprehensive Framework to Implement Debiasing and Calibration Simultaneously in Recommender Systems

## Abstract

Accurate prediction of rates (e.g., click-through rate and conversion rate) and values (e.g., watch time and pay amount) is a fundamental pursuit of modern recommender systems. Due to training sample bias and model training error, few online models are able to deliver an absolutely precise prediction which can fully align with the ideal data distribution. Existing research has developed two technical approaches, i.e., debiasing and calibration, which address training sample bias and model training error respectively, failing to optimize both types of errors simultaneously. In this paper, we propose the **A**daptive **T**raffic **Red**istribution (ATRD) framework, which implements debiasing and calibration from a comprehensive perspective. Firstly, we propose parallel sampling and traffic minimal connected graph to construct a series of comparable samples of item traffic proportion and the corresponding efficiency. Secondly, we fit the function which maps item traffic proportion to its efficiency and solve for the primal traffic proportion. Thirdly, we apply step exploration and subgradient descent to derive the correction factor for online traffic adjustment. Theoretically, the proposed framework can ensure that the exposure of items is more commensurate with their true efficiency by traffic redistribution, leading to the optimization of recommendation results. Online experiments validate the effectiveness of the proposed method and demonstrate significant improvements in business metrics.

## 1 Introduction

Recommender systems (RS) have witnessed a rapid development over the last decade. Precise predictions of rates, such as click-through rate (CTR) and conversion rate (CVR), and values, such as watch time and pay amount, enable RS to efficiently connect users with relevant items, thereby enhancing user experience and amplifying business revenue (Ricci et al., 2015; Zhang et al., 2019). Within this context, most technical evolutions are devoted to improving the accuracy of model predictions.

Figure 1 shows a complete RS pipeline adopted by almost all industrial applications. The pipeline follows a paradigm composed of four stages: constructing observed samples based on online logs, training a model under a certain hypothetical space, deploying an online service with the well-trained model, and making inference for each user on terms of a specific item (Covington et al., 2016). The original intention of recommendation is to employ optimally trained models based on unbiased samples to accurately indicate user preferences for items. Regrettably, few models can achieve theoretically ideal predictions because of limitations from the following two primary aspects. Firstly, training data fails to represent the unbiased distribution due to the influence of recommendation outcomes and user interactions, which leads to training sample bias (Chen et al., 2023). Secondly, inherent flaws existing in each crucial modeling stages, including sample selection, model assumptions, loss function design and so on, prevent perfect fitting to the observed samples, which is known as model training error (Wang, 2023).

To mitigate these problems, existing research has correspondingly developed two correction approaches: debiasing and calibration. The former aims to align observed samples with the true distribution through techniques such as sampling, inverse propensity scoring, and causality-based

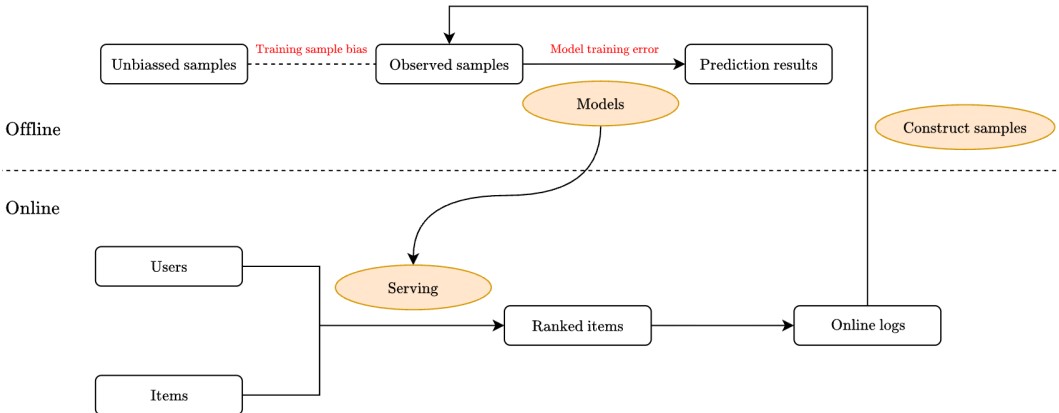

Figure 1: A common pipeline of modern recommender systems.

methods, while the latter seeks to match model predictions with observed sample distributions via methods including training-time and post-hoc calibration (Chen et al., 2023; Wang, 2023). However, both debiasing and calibration only concentrate on eliminating a single error and fail to contribute to a global optimization for the gap between model predictions and unbiased samples.

In this paper, we propose a novel framework, **A**daptive **T**raffic **Red**istribution (ATRD), to address the limitations of existing research from a comprehensive perspective. ATRD focuses on improving recommendation efficiency by dynamically revising traffic proportion for each item, enabling adaptive convergence toward the ideal traffic distribution derived from its true efficiency. ATRD comprises three main steps: firstly we construct a series of comparable samples of item traffic proportion and corresponding efficiency; secondly, we fit a concave function that maps item traffic proportion to its efficiency and obtain the primal traffic proportion by solving for the extreme value of the concave function and step updating; thirdly, we derive the correction factor for online traffic by solving an optimization problem with efficiency as the objective and the primal traffic proportion as constraints.

The principal contributions of this paper are threefold:

- We propose ATRD, a novel framework that simultaneously implements debiasing and calibration, thereby addressing a critical research gap in error correction.
- We present a comprehensive design and implementation of ATRD, detailing its operational mechanics and integration protocol.
- Our proposed framework has been successfully deployed in large-scale business systems, where rigorous online A/B testing validated its effectiveness and superiority.

## 2 RELATED WORKS

**Debiasing**. Sample bias constitutes a pervasive phenomenon in RS. As surveyed in Chen et al. (2023), three primary types of bias notably impact RS: exposure bias arising from training data only composed of exposed items with non-exposed items systematically absent, selection bias reflecting the tendency of users to interact primarily with items matching their interests while overlooking alternatives, and position bias where higher-ranked items are easier to be noticed regardless of relevance. A naive solution mitigates the sample bias through sampling or weight adjustment. Yu et al. (2017) performs negative sampling for popular items, while Ding et al. (2018; 2019) leverage viewed-but-unclicked data to estimate the user exposure as auxiliary sampling information. Chen et al. (2019a) proposes a method that constructs recommendation models using the social network data and proposes social network random walks for negative sampling. Complementary approaches proposed by He et al. (2016); Li et al. (2010) specifically assign weights to unexposed items. However, these methods exhibit excessive dependence on prior knowledge and consequently struggle to determine unbiased sample weights. Meanwhile, several studies have attempted to mitigate sam-

ple bias using Inverse Propensity Scoring (IPS). Works such as Schnabel et al. (2016); Wang et al. (2021) apply inverse propensity weights to data samples, theoretically enabling unbiased estimation. However, accurately estimating propensity scores poses significant challenges, and the inversion of propensity often incurs high variance. Subsequent research has focused on variance reduction techniques, though these solutions warrant further refinement, where Wang et al. (2019) combines data imputation with propensity scoring to propose a doubly robust estimator, Saito (2020) develops an unbiased pairwise learning method, and Zhu et al. (2020) employs joint learning approaches to stabilize propensity score variance. Additionally, Guo et al. (2019); Huang et al. (2021) explicitly model the position bias within the model architecture, though the approach exclusively addresses positional effects. Concurrently, causality-based methods are widely adopted in RS. Xu et al. (2023) incorporates user exposure to items into collaborative filtering, modeling exposure as a latent variable. Yang et al. Yang et al. (2021) mitigate exposure bias through counterfactual sampling, while Zhu et al. (2023) introduces an auxiliary network architecture to generate and learn such counterfactual samples. Complementarily, Zhang et al. (2021) proposes the PDA framework to eliminate the confounding effects of popularity bias while preserving its beneficial components. Nevertheless, these causality-based approaches face inherent limitations, where the absence of ground truth for counterfactual samples impedes reliable evaluation of their effectiveness.

**Calibration**. Model training errors remain inherently unavoidable due to suboptimal hypothesis spaces and immature optimization techniques. Substantial research has pursued model score calibration through two primary methodologies, training-time calibration and post-hoc calibration (Wang, 2023). For training-time calibration, Muttenthaler et al. (2024) proposes odd-k-out learning that minimizes cross-entropy error over sample sets to achieve superior calibration. Han et al. (2024) introduces Dynamic Regularization (DReg) to calibrate model scores during optimization. Yoon et al. (2024) develops a tuning-free trainable objective that intrinsically learns calibration errors. Post-hoc calibration is a simpler and more prevalent category. Some methods calibrate model predictions through parametric scaling such as Gaussian calibration, Beta calibration, Gamma calibration, platt scaling and temperature scaling (Balanya et al., 2024; Kull et al., 2019). Kumar et al. (2019) proposes a hybrid approach which implements a calibrator combing scaling and binning. Gupta & Ramdas (2023) proposes online platt scaling algorithm which combines the platt scaling technique with online logistic regression.

However, due to the persistent presence of sample bias, exclusively optimizing model training errors through calibration often yields suboptimal business outcomes in industrial deployment scenarios. The ATRD framework proposed in this work simultaneously implements debiasing and calibration, thereby holistically addressing the limitations in existing research.

## 3 METHODOLOGY

### 3.1 PROBLEM FORMULATION

Given a user set $I$ and an item set $J$, a recommendation task is to maximize expected efficiency by recommending the most relevant item $j$ to user $i$, which can be formulated as:

$$\max \frac{1}{||I||} \sum_{i \in I} \sum_{j \in J} f_{ij} x_{ij}$$
$$s.t. \sum_{j \in J} x_{ij} = 1 \quad \forall i \in I \tag{1}$$
$$x_{ij} \in [0,1] \quad \forall i \in I, j \in J$$

where $f_{ij}$ denotes the expected utility derived from user $i$ and item $j$, and $x_{ij}$ indicates whether item $j$ is exposed to user $i$. In the setting of large-scale industrial applications, $x_{ij}$ is usually relaxed to a continuous variable for convenience in solving. Obviously, the optimal solution to problem 1 is:

$$x_{ij} = \begin{cases} 1, & j = \arg\max_{j' \in J} f_{ij'} \\ 0, & \text{otherwise} \end{cases}. \tag{2}$$

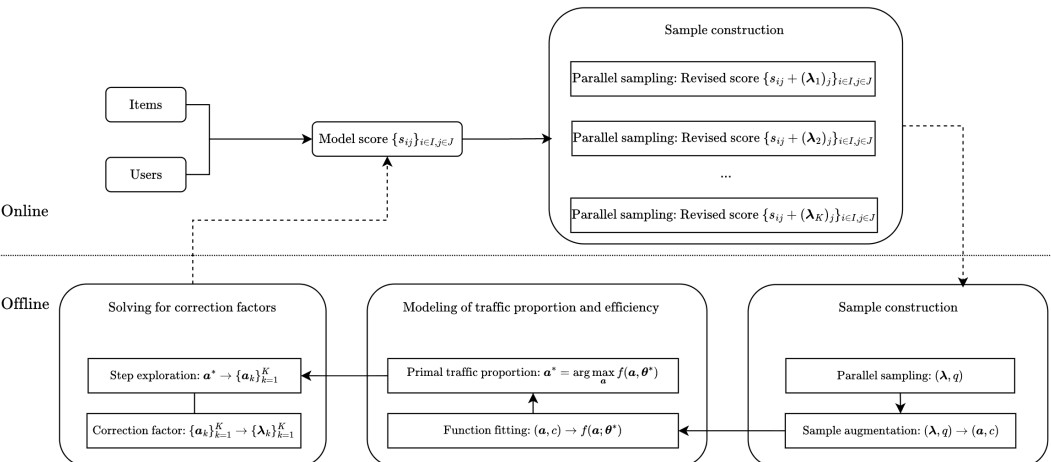

Figure 2: The workflow of ATRD.

Over the last decades, various tree-based and DNN-based models are elaborately designed to predict $f_{ij}$. Denoting $s_{ij}$ as the model prediction for $f_{ij}$, solution 2 can be practically written as

$$x_{ij} = \begin{cases} 1, & j = \arg\max_{j' \in J} s_{ij'} \\ 0, & \text{otherwise} \end{cases}.$$ (3)

Solution 2 and solution 3 are mathematically equivalent if $s_{ij} = f_{ij}$. However, this equivalence can be rarely satisfied in real-world scenarios as discussed above. The system error between $s_{ij}$ and $f_{ij}$, denoted as $y_{ij} = f_{ij} - s_{ij}$, is crucial to addressing this discrepancy. In a large-scale system with millions of users interacting with different items, directly modeling $y_{ij}$ states an enormous challenge. Considering that the number of users often greatly exceeds the number of items, we approximate the system error $y_{ij}$ to a per-item correction factor $\lambda_j$, obtaining an approximate solution as:

$$x_{ij} = \begin{cases} 1, & j = \arg\max_{j' \in J} \{s_{ij} + \lambda_j\} \\ 0, & \text{otherwise} \end{cases}.$$ (4)

Given a correction factor vector $\boldsymbol{\lambda} = [\lambda_1, \lambda_2, ..., \lambda_{||J||}]$, the best traffic allocation strategy is described as solution 4. By applying $\boldsymbol{\lambda}$ online for a period of time, we can compute the traffic efficiency, which is equivalent to the objective of problem 1 . Obviously, the core task is to find a correction factor $\boldsymbol{\lambda}$ and redistribute traffic according to solution 4 to achieve the maximal traffic efficiency.

### 3.2 THE OVERVIEW OF ATRD

The proposed ATRD framework is built around finding a series of correction factor vectors and applying them online to achieve the maximal traffic efficiency. An overview of this framework is illustrated in Figure 2. In a recommender system, the recommendation model rates each item $j$ for each visiting user $i$, yielding a score $s_{ij}$ that decides whether item $j$ is exposed to user $i$. In ATRD, $s_{ij}$ will be intervened by correction factor vectors and then deployed online. The whole process of intervention consists of three stages. The first stage is sample construction, which is to collect online samples of correction factor vectors and corresponding efficiencies, and transfer them into samples of traffic proportion and efficieny coefficient. At the second stage, we build a function to reveal the relationship between traffic proportion and efficieny coefficient, and find the extreme value as primal traffic proportion. The last stage is to solve a series of correction factor vectors according to the primal traffic proportion and intervene in the online model scores. The implementation specifics of these three stages will be detailed in the following sections.

### 3.3 SAMPLE CONSTRUCTION

Given a correction factor vector $\boldsymbol{\lambda}$, we can influence the item traffic proportion over a specific period of time. The efficiency $q$ is computed by collecting statistics on the user exposures and interactions

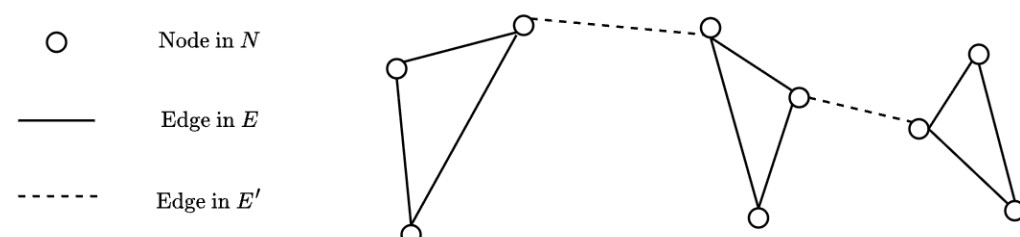

Figure 3: An illustration of T-MCG.

during the period. Obviously, Efficiency $q$ is the actual statistics for the objective of problem 1 with solution 4. Controlling the traffic with different $\boldsymbol{\lambda}$ allows us to enumerate the corresponding online efficiency $q$, thereby identifying the optimal $\boldsymbol{\lambda}$. However, the efficiency $q$ across different time periods is usually incomparable. For instance, the CTR of users between 5:00 AM and 7:00 AM may be significantly lower than that between 5:00 PM and 7:00 PM. We propose two approaches to tackle this challenge:

- Parallel Sampling, which collects pairs of $(\boldsymbol{\lambda}, q)$ as samples in the same period.
- Traffic Minimal Connected Graph (T-MCG), which augments samples in various periods.

### 3.3.1 Sample Collection in the Same Period

Parallel sampling is used online to collect samples from the same period. By setting the number of parallel sampling to $K$, a series of correction factors $\boldsymbol{\lambda}$ are applied simultaneously and intervene in the traffic in a mutually exclusive manner. Each incoming user is intervened by one of these $\boldsymbol{\lambda}$ with equal probability $1/K$. In this period, any two $q_{k_1}$ and $q_{k_2}$ computed based on $\boldsymbol{\lambda}_{k_1}$ and $\boldsymbol{\lambda}_{k_2}$ respectively are comparable as long as the traffic volume is sufficient. Finally, we obtain a group of sample pairs between correction factors and item efficiencies $(\boldsymbol{\lambda}_1, q_1), (\boldsymbol{\lambda}_2, q_2), \ldots, (\boldsymbol{\lambda}_K, q_K)$.

### 3.3.2 Sample Augmentation in Various Time Periods

During distinct periods, various correction factor vectors are deployed to intervene in online traffic, thereby generating different samples. To associate samples with their respective time periods, we attach a time label $t$ to each sample, and represent the samples as $(\boldsymbol{\lambda}, q, t)$. On the one hand, it's impossible to construct a sufficiently large number of samples due to limitations of time and traffic. On the other hand, efficient traffic utilization prohibits exploring an excessive number of samples. It's necessary to take advantage of all samples collected over various periods. Thus, we design an MCG-based algorithm that fully leverages samples from different time periods, called T-MCG, which enables the efficiencies of incomparable sample pairs to be comparable.

Given the user set $I$ and a correction factor vector $\boldsymbol{\lambda}$, the traffic proportion is represented as $\boldsymbol{a} = [a_1, a_2, \ldots, a_{||J||}]$, where $a_j = \frac{1}{||I||} \sum_{i \in I} x_{ij}$ according to solution 4. Thus, sample $(\boldsymbol{\lambda}, q, t)$ can be equivalently transformed into $(\boldsymbol{a}, q, t)$. Let $S$ be the set of all collected samples, where each sample can be written as $s = (\boldsymbol{a}_s, q_s, t_s) \in S$. We define the node set $N = \{\boldsymbol{a}_s | s \in S\}$ to represent the set of explored traffic proportions. Assume there are two samples $m, n \in S$ such that $t_m = t_n$, we construct an edge between node $\boldsymbol{a}_m$ and $\boldsymbol{a}_n$, defined as $e = (\boldsymbol{a}_m, \boldsymbol{a}_n)$. Enumerating all such edges forms a set $E$, and we can define the graph $G = (N, E)$. Notably, if $G$ is a connected graph, it indicates that the efficiencies of all samples are comparable. However, since $G$ is typically an unconnected graph, we need to connect nodes within a sufficient close distance to transform it into a connected graph, where we define the distance between nodes $\boldsymbol{a}_m$ and $\boldsymbol{a}_n$ as $d_{\boldsymbol{a}_m \boldsymbol{a}_n} = ||\boldsymbol{a}_m - \boldsymbol{a}_n||_2$. This task can be formulated as an MCG problem where $d_{\boldsymbol{a}_m \boldsymbol{a}_n}$ denotes the edge cost, and it can be solved via Kruskal's or Prim's algorithm (Ayegba et al., 2020). Let $E'$ denote the set of edges newly added by the algorithm, then the connected graph can be defined as $G' = (N, E \cup E')$. Figure 3 shows a case of $G'$ with $K = 3$.

After constructing graph $G'$, we define efficiency coefficient $c_{\boldsymbol{a}}$ for each node $\boldsymbol{a} \in N$ by fitting the ratio $c_{\boldsymbol{a}_m}/c_{\boldsymbol{a}_n}$ to the efficiency relationship $q_m/q_n$ between arbitrary samples with $t_m = t_n$.

Furthermore, if $(\boldsymbol{a}_m, \boldsymbol{a}_n)$ constitutes an edge in $E'$, we can set $c_{\boldsymbol{a}_m} = c_{\boldsymbol{a}_n}$ as it indicates relative proximity between $\boldsymbol{a}_m$ and $\boldsymbol{a}_n$. Therefore, we can solve the following optimization problem to compute all $c_{\boldsymbol{a}}$:

$$\min \sum_{m \in S} \sum_{n \in S} \mathbb{I}_{\{m,n\}}(w_m + w_n) \left( \frac{q_m}{q_n} - \frac{c_{\boldsymbol{a}_m}}{c_{\boldsymbol{a}_n}} \right)^2 \tag{5}$$

$$s.t. \ c_{\boldsymbol{a}_m} = c_{\boldsymbol{a}_n} \quad \forall (\boldsymbol{a}_m, \boldsymbol{a}_n) \in E'$$

where $w_m$ represents the weight of sample $m$, which can be configured as needed, and $\mathbb{I}_{\{m,n\}}$ is the indicator function defined as equation 6.

$$\mathbb{I}_{\{m,n\}} = \begin{cases} 1, & t_m = t_n \\ 0, & \text{otherwise} \end{cases}. \tag{6}$$

This enables direct efficiency comparison between any two samples through their respective coefficients. Considering that problem 5 is non-convex, it can be approximated by reformulating it as problem 7:

$$\min \sum_{m \in S} \sum_{n \in S} \mathbb{I}_{\{m,n\}}(w_m + w_n) \left( \ln \frac{q_m}{q_n} - \ln \frac{c_{\boldsymbol{a}_m}}{c_{\boldsymbol{a}_n}} \right)^2 \tag{7}$$

$$s.t. \ c_{\boldsymbol{a}_m} = c_{\boldsymbol{a}_n} \quad \forall (\boldsymbol{a}_m, \boldsymbol{a}_n) \in E'$$

This reformulated problem 7 is a convex optimization problem and the optimal solution of all efficiency coefficients $c_{\boldsymbol{a}}$ can be solved using standard optimization solvers.

## 3.4 MODELING OF TRAFFIC PROPORTION AND EFFICIENCY

In this section, we define function $f(\boldsymbol{a}; \boldsymbol{\theta})$ to reveal the relationship between traffic proportion and efficiency. This function can be fitted using samples from graph $G'$. The optimal $\boldsymbol{\theta}^*$ can be solved as:

$$\boldsymbol{\theta}^* = \arg\min_{\boldsymbol{\theta}} \sum_{\boldsymbol{a} \in N} w_n \left[ f(\boldsymbol{a}; \boldsymbol{\theta}) - c_{\boldsymbol{a}} \right]^2. \tag{8}$$

It's hard to directly solve the problem 8 due to the uncertain expression of $f(\boldsymbol{a}; \boldsymbol{\theta})$. Fortunately, we can equivalently transform problem 1 into problem 9, the proof of which can be found in Appendix A.

$$\max \frac{1}{||I||} \sum_{i \in I} \sum_{j \in J} f_{ij} x_{ij}^*(\boldsymbol{a})$$

$$\text{where :}$$

$$\sum_{i \in I} \sum_{j \in J} s_{ij} x_{ij}^*(\boldsymbol{a}) = \max \sum_{i \in I} \sum_{j \in J} s_{ij} x_{ij}$$

$$s.t. \sum_{j \in J} x_{ij} = 1 \quad \forall i \in I \tag{9}$$

$$\sum_{i \in I} x_{ij} = a_j ||I|| \quad \forall j \in J$$

$$\sum_{j \in J} a_j = 1$$

$$x_{ij} \in [0,1] \quad \forall i \in I, j \in J$$

The objective $\frac{1}{||I||} \sum_{i \in I} \sum_{j \in J} f_{ij} x_{ij}^*(\boldsymbol{a})$ of problem 9 represents the efficiency under the given traffic proportion $\boldsymbol{a}$: $\frac{1}{||I||} \sum_{i \in I} \sum_{j \in J} f_{ij} x_{ij}^*(\boldsymbol{a}) = f(\boldsymbol{a}; \boldsymbol{\theta}^*)$, which means solving problem 9 is equivalent to solving the primal traffic proportion: $\boldsymbol{a}^* = \arg\max_{\boldsymbol{a}} f(\boldsymbol{a}; \boldsymbol{\theta}^*)$. As we prove that problem 9 is a convex problem with a concave objective function (found in Appendix B), $f(\boldsymbol{a}; \boldsymbol{\theta})$ is a concave function with respect to $\boldsymbol{a}$. Following the principle of Occam's Razor, we define the efficiency function $f(\boldsymbol{a}; \boldsymbol{\theta})$ as:

$$f(\boldsymbol{a}; \boldsymbol{x}, \boldsymbol{y}, \boldsymbol{z}) = \boldsymbol{a}^2 \boldsymbol{x}^T + \boldsymbol{a} \boldsymbol{y}^T + \mathbf{1} \boldsymbol{z}^T, \text{ where } \boldsymbol{x} < \mathbf{0}. \tag{10}$$

Hence problem 8 can be efficiently solved using algorithms such as BFGS.

## 3.5 SOLVING FOR CORRECTION FACTORS

The efficiency of traffic exhibits inherent fluctuations. When the traffic experiences significant variations, the primal traffic proportion $\boldsymbol{a}^*$ may fluctuate greatly, leading to instability in online performance. To address this issue, we can control the magnitude of traffic proportion updates by setting a proper step size, and define the optimality and exploration uncertainty metrics to evaluate the rationality of the set step. We use the distance to the optimal traffic proportion to evaluate optimality. Specifically, the optimality value of a traffic proportion $\boldsymbol{a}$ is defined as:

$$g(\boldsymbol{a}) = -|\boldsymbol{a}^* - \boldsymbol{a}|, \tag{11}$$

where a smaller $g(\boldsymbol{a})$ indicates a larger distance between $\boldsymbol{a}^*$ and $\boldsymbol{a}$, and thereby a lower optimality. We assume that neighboring traffic proportions exhibit greater similarity. If a large number of previously explored traffic proportions exist near $\boldsymbol{a}$, its uncertainty is considered small. Based on this assumption, the uncertainty value is defined as:

$$h(\boldsymbol{a}) = \frac{1}{\sum_{\boldsymbol{a}' \in M} \frac{w(\boldsymbol{a}')}{\delta + |\boldsymbol{a} - \boldsymbol{a}'|}}, \tag{12}$$

where $\delta$ is the unit length of exploration, and $M$ is the set of explorable traffic proportion. The exploration value of $\boldsymbol{a}$ is then defined as:

$$r(\boldsymbol{a}) = \frac{g(\boldsymbol{a}) - \min g(\boldsymbol{a})}{\max g(\boldsymbol{a}) - \min g(\boldsymbol{a})} + \alpha * \frac{h(\boldsymbol{a}) - \min h(\boldsymbol{a})}{\max h(\boldsymbol{a}) - \min h(\boldsymbol{a})}. \tag{13}$$

Based on $r(\boldsymbol{a})$, we can sample and obtain traffic proportions, represented by $\{\boldsymbol{a}^k\}_{k=1}^K$. It is worth noting that, to reduce the cost of generating the T-MCG, one of the sampled traffic proportions can be deliberately set to be identical to a traffic allocation ratio from the previous time period. By substituting each $\boldsymbol{a}^k \in \{\boldsymbol{a}^k\}_{k=1}^K$ into the subproblem of problem 9, and applying Lagrangian duality theory, we derive its dual problem 14:

$$\min_{\boldsymbol{\lambda}} \max_{x_{ij} \in \mathbb{X}} [\sum_{i \in I} \sum_{j \in J} s_{ij} x_{ij} - \sum_{j \in J} \lambda_j (a_j^k ||I|| - \sum_{i \in I} x_{ij})]$$

$$\text{where: } \mathbb{X} = \{x_{ij} | \sum_{j \in J} x_{ij} = 1, \forall i \in I \text{ and } x_{ij} = \{0, 1\}, \forall i \in I, j \in J\} \tag{14}$$

Problem 14 can be solved using a subgradient algorithm. The detailed logic is provided in Algorithm 1, and its output corresponds to the optimal $\boldsymbol{\lambda}$.

## 3.6 AN APPLICATION ANALYSIS OF ATRD

This framework utilizes online sampled data to revise the traffic proportion for items, with the goal of maximizing traffic efficiency, theoretically making it applicable to all recommendation problems. In practical applications, the applicability of this framework is influenced by two factors: On the one hand, when the number of items is large, the relationship between traffic proportion and efficiency becomes more complex, necessitating a larger number of samples to fit a more stable function. However, the training samples for fitting the function are obtained from online sampling and cannot be accumulated in large quantities within a short period of time. On the other hand, although this framework leverages the traffic allocation-efficiency function to systematically address the overall error in RS, it introduces inherent noise in the sample efficiency. Consequently, this framework is expected to achieve higher benefits in scenarios where the number of recommended items is relatively small and efficiency metrics are less noisy.

## 4 ONLINE EXPERIMENTS

ATRD is a recommendation framework that leverages online sampled data to provide real-time feedback. Currently, there is no publicly available dataset to validate this framework, and since this approach is directly targeted at optimizing business revenue, precision alone may not fully represent its effectiveness. To demonstrate the practicality and performance of this framework, we conduct a real-world industrial case study. The case study is based on an important recommendation scenario

---

**Algorithm 1:** SubGradient Descent for $\boldsymbol{\lambda}$

---

**Input**: Initial dual variable $\boldsymbol{\lambda}$
**Output**: The optimal $\boldsymbol{\lambda}$

---

Initialize iteration counter $t \leftarrow 0$
**repeat**

Update $x_{ij}^*$:
$$x_{ij}^* = \begin{cases} 1, & j = \arg\max_j\{\lambda_j + s_{ij}\} \\ 0, & \text{otherwise} \end{cases}$$

Compute gradient:
$$\boldsymbol{g}(\boldsymbol{\lambda}) = ||I||\boldsymbol{a}^k - \sum_{i \in I} \boldsymbol{x}_i^*$$

Choose step size $l_t$ such that:
$$l_t \to 0, \quad \sum_{t=1}^{\infty} l_t = +\infty$$

Update dual variable:
$$\boldsymbol{\lambda} = \max(\boldsymbol{\lambda} - l_t\boldsymbol{g}(\boldsymbol{\lambda}), \boldsymbol{0})$$

Increment $t$:
$$t \leftarrow t + 1$$

**until** $\boldsymbol{\lambda}$ *converges*;

---

Table 1: Data description of the experimental scenario.

| Item | Exposures | Conversion Amount per Unit |
|---|---|---|
| List A | 10,357 | 10 |
| List B | 6,221 | 13 |

in an online financial platform. In this scenario, two lists are available: List A and List B, where the task is to prioritize displaying one of them to users. Users can click and switch to view the other list. The optimization goal of ATRD is to decide which list should be shown first to the user to maximize the business revenue. Typically, a model can be trained with conversion amounts as labels, and the prioritized list is determined based on the model scores. However, the data from this scenario (as shown in Table 1) indicate that while List A has the largest exposure, its per-unit efficiency is the lowest. Training a model based on this data may result in higher scores for List B. In fact, most users first see List A by default upon entering the platform, which leads to a lower observed conversion efficiency for List A. Conversely, List B is predominantly exposed only after users actively click on it, resulting in a higher observed conversion efficiency. Thus, relying solely on the scores generated by the recommendation model may misjudge List A as being less efficient than List B, which contradicts the actual efficiency.

We conducted extensive experiments in this scenario, with GMV (Gross Merchandise Value) as the optimization objective. The baseline bucket used the recommendation model to determine the priority of the lists. The baseline model was constructed with a BST + ESMM architecture and trained on the last year of historical data from this scenario (Chen et al., 2019b; Ma et al., 2018). The experimental bucket adopted the ATRD, determining whether List A or List B should be prioritized as the decision problem. The algorithm performed three rounds of parallel sampling and triggered updates after a fixed number of exposures. During the experiment, the cumulative conversion amount achieved a statistically significant improvement of 13%, verified by T-test with 95% confidence intervals. As shown in Figure 4, the proportion of prioritized exposures for List A in the baseline bucket remained stable at approximately 30%, while the proportion in the ATRD bucket increased amid fluctuations and eventually stabilized at around 60%. In the baseline bucket, List A has 48% lower exposure than List B, but its efficiency in terms of conversion amount was 27% higher. In contrast, in the ATRD bucket, List A was exposed more frequently than List B, with 22% higher exposure, while their conversion efficiency differed only by a small margin of 8%. This validates

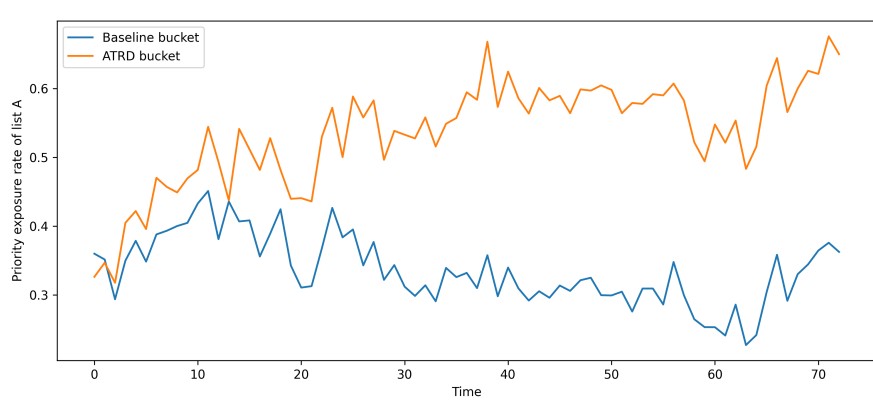

Figure 4: Priority exposure rate of List A in baseline and ATRD bucket.

that efficiency changes alongside traffic proportion, suggesting that sample data may contain "misleading information". As expected, the higher exposure of List B in the baseline bucket resulted in greater conversion amounts for List B compared to the ATRD bucket. Conversely, the ATRD bucket achieved higher conversion amounts for List A, which exceeded the deficit observed in List B's conversion amounts. This demonstrates that the adaptive traffic optimization algorithm can iteratively explore historical information to identify "misleading information", correct the scoring errors caused by such information, and thereby improve online conversion amounts.

## 5 CONCLUSION

In this paper, we propose a novel framework, Adaptive Traffic Redistribution (ATRD), to address the limitations of existing research on debiasing and calibration. We establish a systematic pipeline, which can be seamlessly embedded into any recommender systems no matter what model provides service. Theoretically, ATRD can eliminate training sample bias and model training error simultaneously, and online A/B test also demonstrates its effectiveness and superiority. Further works will concentrate on exploration of ATRD's application in large-scale recommender systems.

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

## A  APPENDIX

**Lemma 1**. Problem 1 is equivalent to problem 15.

$$
\max \sum_{i \in I} \sum_{j \in J} f_{ij} x_{ij}
$$
$$
s.t. \sum_{j \in J} x_{ij} = 1 \quad \forall i \in I
$$
$$
\sum_{i \in I} x_{ij} = a_j \|I\| \quad \forall j \in J \tag{15}
$$
$$
\sum_{j \in J} a_j = 1
$$
$$
x_{ij} \in [0,1] \quad \forall i \in I, j \in J
$$
$$
a_j \in [0,1] \quad \forall j \in J
$$

Let the feasible region of problem 1 and problem 15 be $\mathbb{X}_1$ and $\mathbb{X}_2$ respectively. For $\forall \boldsymbol{x} \in \mathbb{X}_2$, $\sum_{j \in J} x_{ij} = 1$, which implies $\mathbb{X}_2 \subset \mathbb{X}_1$.

For $\forall \boldsymbol{x} \in \mathbb{X}_1$, we have:

$$\sum_{j \in J} x_{ij} = 1.$$

$$\sum_{i \in I} x_{ij} = a_j \|I\| \implies a_j = \frac{\sum_{i \in I} x_{ij}}{\|I\|}.$$

Since

$$\sum_{i \in I} \sum_{j \in J} x_{ij} = \|I\| \text{ and } x_{ij} \in [0, 1] \implies a_j \in [0, 1],$$

for $\forall \boldsymbol{x} \in \mathbb{X}_1$, we can always find the variable

$$a_j = \frac{\sum_{i \in I} x_{ij}}{\|I\|}$$

such that

$$a_j \in [0, 1] \text{ and } \sum_{i \in I} x_{ij} = a_j \|I\|,$$

which implies $\mathbb{X}_1 \subset \mathbb{X}_2$.

By combining the above, we have $\mathbb{X}_1 = \mathbb{X}_2$. Since the objective functions of problem 1 and problem 15 are identical.

**Lemma 2**. Problem 15 is equivalent to problem 9.

Under the assumption that $\lambda_j = f_{ij} - s_{ij}$, for $\forall \boldsymbol{a} \in \{\mathcal{R}^{|J|} \cap \sum_{j \in J} a_j = 1\}$, we have:

$$\sum_{i \in I} \sum_{j \in J} s_{ij} x_{ij} = \sum_{i \in I} \sum_{j \in J} f_{ij} x_{ij} - \sum_{j \in J} \left( \lambda_j \sum_{i \in I} x_{ij} \right) = \sum_{i \in I} \sum_{j \in J} f_{ij} x_{ij} - \sum_{j \in J} \lambda_j a_j \|I\|.$$

Here, $\sum_{j \in J} \lambda_j a_j \|I\|$ is constant with respect to $x_{ij}$. Therefore, the subproblems of problem 15 and problem 9 have equivalent optimal solutions. Let $x_{ij}^*(\boldsymbol{a})$ denote the optimal solution of problem 15 given $\boldsymbol{a}$. Then $x_{ij}^*(\boldsymbol{a})$ is also the optimal solution to the subproblem of problem 9. Substituting $x_{ij}^*(\boldsymbol{a})$ into the objective functions of both problems confirms that the objective function of the equivalent problem 9 is equal to that of the equivalent problem 15.

Thus, problem 1 is equivalent to problem 9 based on Lemma 1 and Lemma 2.

## B APPENDIX

Let

$$\mathbb{F}(\boldsymbol{a}) = \{x_{ij} \mid \sum_{i \in I} x_{ij} = a_j \|I\|, \sum_{j \in J} x_{ij} = 1, x_{ij} \in [0, 1]\},$$

and suppose the optimal value of subproblem 9 is $z(\boldsymbol{a})$. For

$$\forall \boldsymbol{a}_1, \boldsymbol{a}_2 \in \mathcal{R}^{|J|} \cap \{\boldsymbol{a} \mid \sum_{j \in J} a_j = 1, a_j \in [0, 1]\}.$$

Let $\boldsymbol{a}_3 = \alpha \boldsymbol{a}_1 + (1 - \alpha) \boldsymbol{a}_2$ with $0 \leq \alpha \leq 1$, we have:

$$\alpha z(\boldsymbol{a}_1) + (1 - \alpha) z(\boldsymbol{a}_2) = \alpha \max_{x_{ij} \in \mathbb{F}(\boldsymbol{a}_1)} \sum_{i \in I} \sum_{j \in J} s_{ij} x_{ij} + (1 - \alpha) \max_{x_{ij} \in \mathbb{F}(\boldsymbol{a}_2)} \sum_{i \in I} \sum_{j \in J} s_{ij} x_{ij}$$

$$= \alpha \sum_{i \in I} \sum_{j \in J} s_{ij} x_{ij}^*(\boldsymbol{a}_1) + (1 - \alpha) \sum_{i \in I} \sum_{j \in J} s_{ij} x_{ij}^*(\boldsymbol{a}_2).$$

Let $x_{ij}^3 = \alpha x_{ij}^*(\boldsymbol{a}_1) + (1-\alpha)x_{ij}^*(\boldsymbol{a}_2)$, we have:

1. $\displaystyle\sum_{i\in I} x_{ij}^3 = \sum_{i\in I}[\alpha x_{ij}^*(\boldsymbol{a}_1) + (1-\alpha)x_{ij}^*(\boldsymbol{a}_2)] = \alpha(\boldsymbol{a}_1)_j||I|| + (1-\alpha)(\boldsymbol{a}_2)_j||I|| = (\boldsymbol{a}_3)_j||I||$

2. $\displaystyle\sum_{j\in J} x_{ij}^3 = \sum_{j\in J}[\alpha x_{ij}^*(\boldsymbol{a}_1) + (1-\alpha)x_{ij}^*(\boldsymbol{a}_2)] = 1$

3. $x_{ij}^3 \in [0,1]$

Thus, $x_{ij}^3 \in \mathbb{F}(\boldsymbol{a}_3)$, which means $x_{ij}^3$ is a feasible solution for the subproblem of problem 9.

$$\alpha z(\boldsymbol{a_1}) + (1-\alpha)z(\boldsymbol{a_2}) = \sum_{i\in I}\sum_{j\in J} s_{ij}x_{ij}^3$$

$$\leq \max_{x_{ij}\in\mathbb{F}(\boldsymbol{a}_3)}\sum_{i\in I}\sum_{j\in J} s_{ij}x_{ij}$$

$$= z(\boldsymbol{a}_3)$$

Hence, problem 9 can be expressed as:

$$\max\sum_{i\in I}\sum_{j\in J} f_{ij}x_{ij}^*(\boldsymbol{a}_3) = \max\sum_{i\in I}\sum_{j\in J}(s_{ij}+\lambda_j)x_{ij}^*(\boldsymbol{a}_3)$$

$$= \max\sum_{i\in I}\sum_{j\in J} s_{ij}x_{ij}^*(\boldsymbol{a}_3) + ||I||\sum_{j\in J}\lambda_j(\boldsymbol{a}_3)_j$$

$$= \max\{z(\boldsymbol{a}_3) + ||I||\boldsymbol{\lambda}\boldsymbol{a}_3\}$$

$$\geq \max\{\alpha z(\boldsymbol{a}_1) + (1-\alpha)z(\boldsymbol{a}_2) + \alpha||I||\boldsymbol{\lambda}\boldsymbol{a}_1 + (1-\alpha)||I||\boldsymbol{\lambda}\boldsymbol{a}_2\}$$

$$= \max\{\alpha[z(\boldsymbol{a}_1) + ||I||\boldsymbol{\lambda}\boldsymbol{a}_1]\} + \max\{(1-\alpha)[z(\boldsymbol{a}_2) + ||I||\boldsymbol{\lambda}\boldsymbol{a}_2]\}$$

$$= \alpha\max\sum_{i\in I}\sum_{j\in J} f_{ij}x_{ij}^*(\boldsymbol{a}_1) + (1-\alpha)\max\sum_{i\in I}\sum_{j\in J} f_{ij}x_{ij}^*(\boldsymbol{a}_2)$$

Problem 9 is a convex optimization problem, and its objective function is a concave function with respect to $\boldsymbol{a}$.

