# OpenReview forum: "ATRD: A Comprehensive Framework to Implement Debiasing and Calibration Simultaneously in Recommender Systems"
_ICLR.cc/2026/Conference — Submitted to ICLR 2026_

### Official Review · Reviewer_GJXd · 2025-10-30

**Soundness:** 3
**Presentation:** 3
**Contribution:** 3
**Rating:** 8
**Confidence:** 2

**Summary:**

The  author of ATRD introduces a framework to address two fundamental challenges in modern RS: training sample bias and model training error. The authors argue that existing techniques like debiasing (which corrects for biased training data) and calibration (which corrects for model prediction errors) are typically applied in isolation, failing to solve the overall problem holistically.

The proposed solution is the Adaptive Traffic Redistribution (ATRD) framework, which aims to improve recommendation quality by dynamically adjusting the online traffic allocated to each item, ensuring that an item's exposure is more aligned with its true value or "efficiency."

Key contributions:

* the framework above and how it could be intergrated into a large industrial scale system
* It validates the framework's effectiveness through large-scale online A/B testing in a production environment, demonstrating significant improvements in business metrics

**Strengths:**

originality:

* The key original contribution is the framing of the problem. Instead of treating training sample bias (debiasing) and model training error (calibration) as separate issues to be fixed independently, the authors propose a unified framework to address both simultaneously. This shifts the focus from "fixing the data" or "fixing the model" to dynamically "fixing the traffic allocation," which is a pragmatic and original approach for industrial systems.
* The introduction of the Traffic Minimal Connected Graph (T-MCG) is a creative solution to a difficult practical problem: how to make efficiency metrics from different time periods comparable

Quality:
* The paper's standout strength is its move beyond offline simulation to deployment in a live, large-scale system. The authors present results from a rigorous online A/B test on a financial platform, demonstrating a statistically significant 13% improvement in Gross Merchandise Value

* The analysis of the online experiment is insightful. The paper shows not just that the method worked, but why. By analyzing the traffic proportions before and after ATRD was applied

Clarity: The paper is well-written, clearly structured.

Significance:
The paper tackles the critical gap between offline model performance and online business outcomes. By creating a closed-loop system that uses real-time feedback to correct for both data and model deficiencies, ATRD provides a robust solution to a problem that plagues virtually all large-scale recommendation platforms.

**Weaknesses:**

* generalizability beyond a single domain: The framework defines item efficiency as the ratio of conversions to exposures, which is essentially a conversion rate. This metric is well-suited for the e-commerce or financial domain of the experiment, but it is too narrow for many other recommendation scenarios (e.g., news, video, music streaming) where objectives include long-term engagement, user satisfaction, or exposure to diverse content. Optimizing solely for a simple conversion rate could inadvertently create feedback loops that promote clickbait or narrowly popular items at the expense of user experience. The paper would be more robust if it discussed the limitations of this efficiency metric. The authors could suggest how the ATRD framework might be adapted for more complex, multi-faceted objectives, for instance by incorporating session-based metrics, fairness constraints, or a proxy for long-term user value into the definition of efficiency.


* The paper's central claim is that ATRD implements debiasing and calibration simultaneously and is superior to methods that handle them separately. However, the online experiment only compares ATRD against a "Base" production system. It does not include a baseline where a state-of-the-art debiasing method (e.g., inverse propensity scoring) and a state-of-the-art calibration method (e.g., temperature scaling) are applied in sequence. Without this comparison, it is difficult to definitively conclude that the proposed unified approach is superior to a strong, modular baseline.

* To improve reproducibility, the authors should provide more concrete details about the sample construction process. The paper states that it "collects historical data and constructs a series of comparable samples," but lacks specifics on how these samples are created (e.g., are they from different time windows, from arms of previous A/B tests?). The quality of the fitted concave function is highly dependent on the properties of these samples. Clarifying the granularity of a "sample," the duration of the data collection period, and any criteria used for sample selection would be a valuable addition, perhaps in an appendix.

**Questions:**

A core component of ATRD is the approximation of the complex, per-user, per-item system error (y_{ij} with a single, per-item correction factor (λj). While this makes the problem tractable at scale, it is a strong assumption that averages out all personalization effects. The optimal exposure for an item is often highly dependent on the user. For instance, an item that is highly efficient for one user segment might be inefficient for another. The current formulation cannot capture this nuance. A valuable direction for future work would be to explore contextualized correction factors. For example, could the framework learn separate factors for different user segments to re-introduce a degree of personalization into the traffic redistribution?

---

### Official Review · Reviewer_MrcF · 2025-10-30

**Soundness:** 3
**Presentation:** 3
**Contribution:** 3
**Rating:** 4
**Confidence:** 3

**Summary:**

This paper proposes the Adaptive Traffic Redistribution (ATRD) framework, aiming to jointly optimize "debiasing" and "calibration" in recommender systems by dynamically adjusting item traffic proportions to match their true efficiency. It has certain value in both theoretical design and industrial practice, but the sufficiency of experimental verification needs further improvement.

**Strengths:**

1.Motivation: Aligns with Research Gaps and Is Highly Targeted.The motivation of the ATRD framework is clear and significant. It not only identifies the core problem in existing recommender systems—the coexistence of training sample bias and model training error—but also points out the limitation of existing methods, which only optimize one type of error independently. Focusing on the underaddressed research gap of "implementing debiasing and calibration simultaneously", the framework proposes to optimize both types of errors through traffic redistribution, which is closely aligned with the core needs of practical recommender system applications. This fully meets the requirement of "pointing out the limitations of existing methods for existing problems", making it a notable strength.
2.Technical Soundness: Rational Design to Effectively Address Challenges.The technical route of the framework is complete and logically consistent, enabling it to address the two proposed challenges in a targeted manner. For training sample bias, comparable samples are constructed through parallel sampling and Traffic Minimal Connected Graph (T-MCG) to eliminate the incomparability of efficiency across different time periods, thereby achieving debiasing. For model training error, a concave function mapping traffic proportion to efficiency is fitted to solve the primal traffic proportion, and correction factors are derived using step exploration and subgradient descent to complete calibration. The entire process forms a closed loop from sample construction and function modeling to correction factor solving, with detailed technical details (e.g., using Kruskal/Prim algorithms for T-MCG construction and BFGS algorithm for function fitting). This effectively supports the core goal of simultaneous debiasing and calibration, demonstrating strong technical feasibility.

**Weaknesses:**

1. Insufficient Depth and Breadth, Leading to Limited Challenge.Although the proposed goal of "solving debiasing and calibration simultaneously" is meaningful, the exploration of the problem’s challenge is insufficient. On one hand, the paper fails to fully analyze the coupling relationship between the two types of errors (training sample bias and model training error) and the conflicts that may arise during simultaneous optimization, treating them merely as independent problems to be solved in combination. On the other hand, it does not expand on the challenge of the problem’s applicable boundaries (e.g., computational pressure in scenarios with a large number of items, stability in dynamic traffic scenarios). It only briefly mentions limitations in applicable scenarios without transforming them into core challenges to be overcome, resulting in insufficient prominence of the problem’s challenge.
2. Incomplete Experiment. The experimental design fails to meet the requirements of "key parameter analysis, ablation experiments of key modules, and complexity comparison", resulting in limited supporting strength. First, it does not analyze the impact of key parameters (e.g., the number of parallel sampling K, step size) on experimental results, making it impossible to verify the rationality of parameter settings. Second, no ablation experiments are conducted on core modules such as T-MCG, function fitting, and correction factor solving, so the contribution of individual modules cannot be clarified. Third, it does not compare the time/space complexity between ATRD and the baseline model; although it emphasizes efficiency optimization, there is a lack of quantitative complexity evidence. Furthermore, the experiment is only based on data from two lists in a single financial scenario, with a single scenario and limited data scale, leading to insufficient verification of generalization

**Questions:**

1.In multi-item scenarios (e.g., J=1000), how to optimize the graph construction complexity of T-MCG? Are there dimensionality reduction or sampling strategies to reduce the number of nodes?
2.Do the weights of "optimality" and "uncertainty" in the exploration value r(a) need to be dynamically adjusted? Are there general rules for weight setting in different business scenarios?
3.If the true efficiency of items changes abruptly due to factors such as promotional activities, can the function fitting and correction factor update of ATRD quickly respond to such mutations? Are there corresponding robustness mechanisms?

---

### Official Review · Reviewer_sY8D · 2025-11-09

**Soundness:** 3
**Presentation:** 3
**Contribution:** 3
**Rating:** 4
**Confidence:** 3

**Summary:**

This paper proposes a unified recommendation system correction framework—ATRD , which aims to achieve both "debiasing" and "calibration" simultaneously.
The framework consists of three stages:
1.	Construct comparable samples through Parallel Sampling and Traffic Minimal Connected Graph ;
2.	Fit an efficiency curve in the form of a concave function to describe the relationship between traffic proportion and efficiency;
3.	Use step size exploration and subgradient descent to solve the online correction factor and realize real-time traffic adjustment.
The authors deployed ATRD in an actual financial recommendation scenario, achieving a 13% increase in GMV, which verifies its practicality and effectiveness.

**Strengths:**

1.	It features strong theoretical unity, integrating debiasing and calibration into the same optimization framework for the first time; the problem formulation, proofs, and optimization process are comprehensive and rigorous.
2.	The ideas of Traffic Minimal Connected Graph (T-MCG) and concave function fitting are innovative.

**Weaknesses:**

1.	The experimental scale is limited: it has only been validated in a single binary-classification business scenario (List A/B), and lacks generalization testing across multiple scenarios.
2.	The complexity analysis is insufficient, as no evaluation of the computational costs of Traffic Minimal Connected Graph (T-MCG) and gradient updates is provided.

**Questions:**

1.	What is the sensitivity of the step size parameter \(l_t\) to the final convergence and stability?
2.	If the number of items reaches tens of thousands, is the construction and fitting of the Traffic Minimal Connected Graph (T-MCG) scalable?

---

### Official Review · Reviewer_PJvD · 2025-11-10

**Soundness:** 2
**Presentation:** 1
**Contribution:** 1
**Rating:** 2
**Confidence:** 5

**Summary:**

This paper proposes the ATRD (Adaptive Traffic Redistribution) framework, achieving simultaneous optimization of bias removal and calibration in recommendation systems for the first time, thereby filling the gap in existing research that only addresses one type of error. The effectiveness of the ATRD framework was validated through industrial-scale online A/B testing, achieving a 13% increase in GMV in a financial platform scenario, demonstrating its practical deployment value.

**Strengths:**

* S1: Breaking through the limitations of existing research that separates “debiasing” and “calibration,” this paper optimizes training sample bias and model training error from the perspective of flow redistribution.
* S2: Parallel sampling ensures comparability of samples within the same time period. T-MCG achieves cross-period sample expansion through graph connectivity, resolving the critical engineering challenge of sparse and incomparable samples in online recommendation systems.

**Weaknesses:**

* W1: The methods described in the paper and the figure representations are unclear and complicated to read. The experimental section is too brief, and the results are insufficient to demonstrate the method's validity.
* W2: The paper's novelty has been overstated, as the method was only validated in the “two-list priority sorting” scenario on financial platforms. It did not cover multiple scenarios, such as e-commerce or content platforms, nor did it test performance with large item quantities, making it impossible to prove the method's universality.
* W3: As the number of items increases, the relationship between traffic proportion and efficiency becomes more complex, requiring extensive sample-based function fitting. However, the paper does not explain how to accumulate sufficient samples within a short timeframe, nor does it mention metrics for large-scale deployment, such as graphics memory and throughput.
* W4: The study did not compare with the latest debias methods (e.g., causal inference, IPS-based methods, and Doubly Robust) or calibration methods (e.g., dynamic calibration, post-processing calibration). It also failed to conduct ablation experiments to validate the necessity of each module, leaving the contribution of the core innovation unclear.

**Questions:**

See weakness in details.

---

### Official Review · Reviewer_f8eQ · 2025-11-11

**Soundness:** 2
**Presentation:** 1
**Contribution:** 2
**Rating:** 2
**Confidence:** 3

**Summary:**

This paper proposes the ATRD framework, which aims to simultaneously address training sample bias (debiasing) and model training error (calibration) in recommender systems through adaptive traffic redistribution to optimize efficiency.

**Strengths:**

The paper's Online A/B tests preliminarily show positive effects on business metrics (GMV).

**Weaknesses:**

(1) The innovation is incremental. Debiasing and calibration are well-established fields in recommendation, and combining them offers limited novelty.

(2) The presentation quality is poor. The text and figures are crude, vague, and contain typos (e.g., "unbiassed" In Figure 1).

(3) The authors' claim that existing debiasing and calibration methods fail at "global optimization" for the gap between model predictions and unbiased samples. I argue that many existing debias methods are designed to achieve this.

**Questions:**

(1) The definition of "Efficiency" is ambiguous. Does it refer to CVR, GMV, or another specific metric?

(2) In the online experiment, why does the paper only compare against a "standard industrial" baseline? It seems to lack comparison with other state-of-the-art debiasing or calibration algorithms.

---

### Meta-Review · Area_Chair_6XFM · 2026-01-06

**Summary:**

Reviewers noted the following concerns.
- Novelty is limited. Combining debiasing and calibration, both well-established in the field, offers incremental contribution.
- Presentation can be significantly improved, including its clarity and unpolished writing, and more elaboration on technical details.
- Experimental results could be improved with comparison to more recent debiasing methods or calibration methods.
- Lack of discussion when number of items are large in real-world deployment

**Reviewer Concerns:**

Authors did not provide rebuttal.

**Reviewer Scores:**

Authors did not provide rebuttal.

---

### Decision · Program_Chairs · 2026-01-26

Reject